# Correlation of the disease-specific Canadian Cardiovascular Society (CCS) classification and health-related quality of life (15D) in coronary artery disease patients

Jarno Kotajärvi[1], Anna-Maija Tolppanen[2,3], Juha Hartikainen[4,5], Heikki Miettinen[4], Marketta Viljakainen[4], Janne Martikainen[3], Risto P. Roine[1,6], Piia Lavikainen[3,4]*

1 Department of Health and Social Management, University of Eastern Finland, Kuopio, Finland, 2 Kuopio Research Centre of Geriatric Care, University of Eastern Finland, Kuopio, Finland, 3 School of Pharmacy, University of Eastern Finland, Kuopio, Finland, 4 Heart Center, Kuopio University Hospital, Kuopio, Finland, 5 School of Medicine, University of Eastern Finland, Kuopio, Finland, 6 Kuopio University Hospital, Kuopio, Finland

* piia.lavikainen@uef.fi

**Data Availability Statement:** Access to data is regulated by the European Union and Finnish laws.

## Abstract

### Background

Generic health-related quality of life (HRQoL) and disease-specific instruments measure HRQoL from different aspects, although generic instruments often contain dimensions that reflect common symptoms. We evaluated how the change in 15D HRQoL and Canadian Cardiovascular Society (CCS) grading of angina severity correlate among coronary artery disease patients during 12-month follow-up.

### Methods

Altogether 1 271 patients scheduled for coronary angiography between June 2015 and February 2017 returned the 15D HRQoL and CCS questionnaires before angiography and after one-year follow-up as a part of routine clinical practice. Spearman correlations between one-year changes in the CCS and the 15D and its dimensions were evaluated. Changes in 15D were classified into 5 categories based on the reported minimal important difference (MID) for the instrument.

### Results

Change in the CCS grade correlated moderately with the MID-based change in the 15D (r = 0.33, 95% confidence interval 0.27–0.39). Correlations between these instruments were similar in different age groups, between sexes and treatment modalities. Of the individual 15D dimensions, changes in breathing (r = 0.40) and vitality (r = 0.30) had the strongest correlations with CCS change.

Data are available from the Kuopio University Hospital for researchers who meet the criteria as required by the European Union and Finnish laws for access to confidential data. Contact persons who will distribute data upon request to qualified researchers: Professor Juha Hartikainen and Director of Research and Innovations Juha Töyräs, Kuopio University Hospital, PO BOX 100, KYS FI-70029, Finland; juha.hartikainen@kuh.fi and juha.toyras@kuh.fi.

**Funding:** This study received funding from the State Clinical Research Fund (VTR) of Kuopio University Hospital (award number 13.11.2014/19§). The funder provided support in the form of salaries for authors [study nurse MV], but did not have any additional role in the study design, data collection and analysis, decision to publish, or preparation of the manuscript. The specific roles of these authors are articulated in the 'author contributions' section.

**Competing interests:** I have read the journal's policy and the authors of this manuscript have the following competing interests: Prof Martikainen is a founding partner of ESiOR Oy and a board member of Siltana Oy. These companies were not involved in carrying out this research. This does not alter our adherence to PLOS ONE policies on sharing data and materials. No other disclosures were reported.

## Conclusion

The symptom-based evaluation of the change in the CCS grade may not catch the full benefit or harm of the treatment and vice versa, a generic instrument, such as 15D, likely does not fully capture change in disease-specific symptoms. Thus, generic and disease-specific instruments are complementary and should be used in conjunction.

## Introduction

Patient reported outcomes (PROs), such as generic and disease-specific health-related quality of life (HRQoL) instruments or measures of symptom severity, are commonly used to evaluate the effectiveness of treatments. Generic and disease-specific instruments have different assets. Generic HRQoL instruments, such as the 15D [1, 2], provide a wider perspective of an individual's quality of life and can be used for calculation of quality-adjusted life years (QALYs) [3–7]. Disease-specific instruments, such as the Canadian Cardiovascular Society (CCS) [8, 9] grading used to assess symptom severity in coronary artery disease (CAD), are usually more sensitive to changes in the specific symptoms of a disease. However, generic instruments typically contain dimensions, which can reflect disease-specific changes. For example, the 15D instrument includes breathing and discomfort and symptoms which can indicate the severity of CAD.

The CCS grading is the most commonly used tool to evaluate the symptoms of CAD in clinical practise, and it is usually rated by the treating physician [10]. The CCS is easy to assess and it provides a rapid assessment of a patient's symptom severity whereas responses to a HRQoL questionnaire, such as the 15D, needs first to be converted, by using a set of utility or preference weights, to a single health index utility score [2, 11]. As disease-specific measures do not provide similar preference-based quality of life information as generic HRQoL instruments, there have been mapping studies between these two types of measurements [12]. It is important to understand how measured outcomes reflect the patient's overall quality of life, regardless on whether the application area of the results is clinical practice or health economic analysis [13–15].

Previous studies have assessed the correlation between clinician measured CCS grading or New York Heart Association (NYHA) functional classification and patient-reported generic HRQoL, mainly in cross-sectional designs [16–18]. Therefore, it is unknown whether the change in symptom severity evaluated by the CCS is associated with the change in generic HRQoL. Further, the earlier studies have usually compared physician-evaluated CCS to a patient-reported generic HRQoL instrument. Therefore, it is unknow whether the change in self-reported CCS correlates with the change in self-reported generic HRQoL. To evaluate this, we investigated the correlation between the change in patient-reported CCS and the generic 15D HRQoL score among patients with coronary artery disease.

## Material and methods

### Study design and population

This observational cohort study was conducted as a part of routine clinical practice. Data were collected from the Kuopio University Hospital Heart Center. Patients scheduled for coronary angiography between June 2015 and February 2017, and subsequently treated by optimal medical therapy (OMT), coronary artery bypass surgery (CABG) or percutaneous coronary

intervention (PCI) filled in the CCS and 15D HRQoL questionnaires pre-angiography at base-line and after a one-year of follow-up.

The NOMESCO Classification of Surgical Procedures codes [19] were used to identify angina patients. After excluding patients with more than one operation during the same treatment episode, the total population included 1271 patients. Of them, 603 were treated with OMT only, 240 with OMT plus CABG and 428 with OMT plus PCI.

### Informed consent and ethical considerations

The study was approved by the Research Ethics Committee of the Northern Savo Hospital District (number 3/2014). Written informed consent was obtained from all individual participants included in this study.

### Instruments

The CCS was used to define symptom severity [8]. The CCS grading system employs four grades from I (without limitation of physical activity) to IV (inability to carry out any physical activity without discomfort) [8, 9, 20, 21]. The CCS is a widely used instrument and its correlation with mortality has been established previously [22, 23].

The 15D instrument is a generic, patient-reported questionnaire for measuring HRQoL [2]. It consists of 15 dimensions (mobility, vision, hearing, breathing, sleeping, eating, speech, excretion, usual activities, mental function, discomfort and symptoms, depression, distress, vitality, and sexual activity) with five ordinal levels. Based on the dimension scores, a single index score ranging from 0 (being dead) to 1 (being in full health) can be calculated. Missing values in 3 or less dimensions were imputed as described previously [1] and the index score was calculated from the health state descriptive system by utilizing the Finnish utility weights [2].

The change in the CCS grades between the baseline and 12-month follow-up was reported on a scale of 1 to 5, with "1" indicating marked deterioration (12-month grade minus baseline grade $\geq$2), "2" moderate deterioration (12-month grade minus baseline grade 1), "3" no change, "4" moderate improvement (12-month grade minus baseline grade -1) and "5" marked improvement (12-month grade minus baseline grade $\leq$-2). For the 15D total score, the change, i.e. the difference in scores between the baseline and 12-month follow-up, was classified into 5 categories based on the reported minimal important difference (MID) for the instrument [24]. MID represents the smallest change in the HRQoL score that can be considered to be important from a clinical perspective [25]. The change in each 15D dimension was scaled with a similar scale as used for the CCS grades. The used scales of changes are presented in Table 1.

### Statistical analysis

Differences in baseline CCS distributions between treatment groups (OMT, CABG and PCI) were tested with Chi Square test. Correlations between changes in the CCS with changes in the 15D and its dimensions from baseline to one year after entering the treatment were studied using Spearman correlation coefficient, as it is suitable for re-ranked values of processed data [26]. Correlations were examined in the total study population and within treatment subgroups, and age and gender groups. The population was divided into two groups based on baseline age (<70 or $\geq$70 years). Patients who had missing values in more than three 15D dimensions were included in dimension-specific analyses, but excluded from the main analyses as the 15D total score cannot be derived in these cases. All statistical analyses were

**Table 1. Categorisation of changes in CCS and 15D.**

| | Change in the CCS grade | Change in the 15D[a] | Change in 15D dimensions |
|---|---|---|---|
| Definition | 12-month grade minus baseline grade | 12-month index score minus baseline index score | 12-month level minus baseline level |
| Scaled Change | | | |
| 1 = much worse | ≥2, deterioration | <-0.035 | ≥2, deterioration |
| 2 = slightly worse | 1 | -0.035 − -0.015 | 1 |
| 3 = no change | 0, no change | -0.015< − <0.015 | 0, no change |
| 4 = slightly better | -1 | 0.015 − 0.035 | -1 |
| 5 = much better | ≤-2, improvement | >0.035 | ≤-2, improvement |

[a]Based on the reported minimal important difference (MID) for the instrument [24].

conducted using SAS 9.4 (SAS Institute Inc., Cary, NC, USA) and R 3.6.3 (The R Foundation for Statistical Computing, Vienna, Austria).

## Results

### Population characteristics

Population characteristics by the treatment groups are presented in Table 2. The proportion of women differed between the treatment groups (48.1% of the OMT, 16.3% of the CABG and 24.8% of the PCI patients). The mean age was similar in all three treatment groups.

Distributions of baseline CCS grades differed between the treatment groups (p<0.001). Of all the participants, 60.0% reported CCS grade II at the baseline. The mean 15D score was 0.831 (standard deviation [SD] 0.098) at the baseline. The CABG group had a higher baseline score (0.857; SD 0.085) compared to the PCI group (0.832; SD 0.098) and the OMT group (0.820; SD 0.101).

### Changes in the individual HRQoL instruments

A total of 814 (64.0%) patients reported both baseline and 12-month follow-up CCS grade. Of them, 350 (43.0%) patients reported an improvement in the CCS grade (Table 3). This was

**Table 2. Participant characteristics.**

| | OMT (n = 603, 47.4%) | CABG (n = 240, 18.9%) | PCI (n = 428, 33.7%) | Total (N = 1271, 100%) |
|---|---|---|---|---|
| Mean age at baseline (SD) | 67.4 (10.0) | 67.4 (8.8) | 68.8 (9.0) | 67.9 (9.5) |
| Age <70 years, n (%) | 334 (55.4%) | 136 (56.7%) | 223 (52.1%) | 693 (54.5%) |
| Men, n (%) | 313 (51.9%) | 201 (83.8%) | 322 (75.2%) | 836 (65.8%) |
| CCS at baseline | | | | |
| I | 64 (10.6%) | 36 (15.0%) | 29 (6.8%) | 129 (10.1%) |
| II | 332 (55.1%) | 148 (61.7%) | 282 (65.9%) | 762 (60.0%) |
| III | 115 (19.1%) | 48 (20.0%) | 76 (17.8%) | 239 (18.8%) |
| IV | 59 (9.8%) | 6 (2.5%) | 23 (5.4%) | 88 (6.9%) |
| *Missing data* | *33 (5.5%)* | *2 (0.8%)* | *18 (4.2%)* | *53 (4.2%)* |
| Mean change in CCS[a] (95% CI) | -0.39 (-0.48 − -0.30) (n = 365) | -0.64 (-0.75− -0.53) (n = 169) | -0.43 (-0.52 − -0.33) (n = 280) | -0.45 (-0.51 − -0.39) (n = 814) |
| *Missing data* | *236 (39.5%)* | *71 (29.6%)* | *148 (34.6%)* | *457 (36.0%)* |
| Mean 15D at baseline (95% CI) | 0.820 (0.101) (n = 593) | 0.857 (0.085) (n = 238) | 0.832 (0.098) (n = 423) | 0.831 (0.098) (n = 1254) |
| *Missing data* | *10 (1.7%)* | *2 (0.8%)* | *5 (1.2%)* | *17 (1.3%)* |
| Mean change in 15D[a] (95% CI) | -0.004 (-0.010−0.003) (n = 387) | 0.034 (0.020−0.047) (n = 173) | 0.016 (0.009−0.024) (n = 292) | 0.011 (0.006−0.016) (n = 852) |
| *Missing data* | *216 (35.8%)* | *67 (27.9%)* | *136 (31.8%)* | *419 (33.0%)* |

[a]12-months score minus baseline.

**Table 3. Changes in the CCS and 15D from baseline to 12 months follow-up by subgroups.** The values are n (%).

| | Treatment group | | | Age group | | Gender | | Total |
|---|---|---|---|---|---|---|---|---|
| | OMT | CABG | PCI | <70 | ≥70 | male | female | |
| **Change in CCS** | n = 365 | n = 169 | n = 280 | n = 425 | n = 389 | n = 534 | n = 280 | n = 814 (**100%**) |
| (≥2, deterioration) **1** | 5 (1.4%) | 0 (0.0%) | 6 (2.1%) | 6 (1.4%) | 5 (1.3%) | 6 (1.1%) | 5 (1.8%) | **11 (1.4%)** |
| (1) **2** | 22 (6.0%) | 8 (4.8%) | 15 (5.4%) | 25 (5.9%) | 20 (5.1%) | 30 (5.6%) | 15 (5.4%) | **45 (5.5%)** |
| (0, no change) **3** | 208 (57.0%) | 61 (36.1%) | 139 (49.6%) | 200 (47.1%) | 208 (53.5%) | 255 (47.8%) | 153 (54.6%) | **408 (50.1%)** |
| (-1) **4** | 95 (26.0%) | 87 (51.5%) | 99 (35.4%) | 150 (35.3%) | 131 (33.7%) | 199 (37.3%) | 82 (29.3%) | **281 (34.5%)** |
| (≤-2, improvement) **5** | 35 (9.6%) | 13 (7.7%) | 21 (7.5%) | 44 (10.4%) | 25 (6.4%) | 44 (8.2%) | 25 (8.9%) | **69 (8.5%)** |
| *Missing values* | | | | | | | | ***n = 457*** |
| **Change in 15D**[a] | n = 387 | n = 173 | n = 292 | n = 441 | n = 411 | n = 554 | n = 298 | n = 852 (**100%**) |
| (much worse) **1** | 107 (27.6%) | 27 (15.6%) | 58 (19.9%) | 93 (21.1%) | 99 (24.1%) | 128 (23.1%) | 64 (21.5%) | **192 (22.5%)** |
| (slightly worse) **2** | 45 (11.6%) | 13 (7.5%) | 31 (10.6%) | 42 (9.5%) | 47 (11.4%) | 55 (9.9%) | 34 (11.4%) | **89 (10.4%)** |
| (no change) **3** | 87 (22.5%) | 27 (15.6%) | 55 (18.8%) | 94 (21.3%) | 75 (18.2%) | 100 (18.1%) | 69 (23.2%) | **169 (19.8%)** |
| (slightly better) **4** | 54 (14.0%) | 16 (9.2%) | 44 (15.1%) | 48 (10.9%) | 66 (16.1%) | 76 (13.7%) | 38 (12.7%) | **114 (13.4%)** |
| (much better) **5** | 94 (24.3%) | 90 (52.0%) | 104 (35.6%) | 164 (37.2%) | 124 (30.2%) | 195 (35.2%) | 93 (31.2%) | **288 (33.8%)** |
| *Missing values* | | | | | | | | ***n = 419*** |

[a]Based on the reported minimal important difference (MID) for the instrument [24].

observed most often among the CABG treated patients of whom 59.2% experienced relief of their symptoms. Only 56 (6.9%) patients reported worsening of CCS grade. However, 408 (50.1%) of patients did not experience any change in their symptoms.

Of the 852 (67.0%) patients who filled in the 15D questionnaires at both time-points, almost half (47.2%) experienced a clinically important improvement (>0.015) in their 15D score according to the MID of the instrument (Table 3). Compared to the changes in the CCS, where only 6.9% reported deterioration of symptoms during the follow-up, 281 (33.0%) of the patients had a worse (<-0.015) 15D score and only 169 (19.8%) reported no change in their 15D score. Compared to other subgroups, a majority of CABG treated patients (61.3%) reported improved (>0.015) 15D score at the end of the follow-up.

## Correlation between changes in CCS and 15D

Altogether 805 (63.3%) patients had both CCS and 15D index scores reported at both time-points and were included in these analyses. In all subgroups, correlations between the changes in the CCS grade and the MID-based changes in the 15D ranged from weak to only moderate. Among all patients, the Spearman correlation coefficient of the change in the two instruments was 0.33 (95% confidence interval [CI] 0.27–0.39) (Fig 1, S1 Table).

The correlation coefficients between the changes in the CCS grade and the MID-based changes in the 15D were stronger in those treated with PCI (r = 0.39; 95% CI 0.28–0.48) than in CABG (r = 0.29; 95% CI 0.15–0.43) or OMT (r = 0.27; 95% CI 0.18–0.37) treatment groups (Fig 2A, S1 Table). Similarly, correlation coefficients were stronger in men than in women and in younger patients (age <70 years) than in older patients (Fig 2B and 2C, S1 Table).

Of the individual 15D dimensions, the change in the breathing dimension (r = 0.40; 95% CI 0.34–0.45) followed by the vitality (r = 0.30; 95% CI 0.24–0.36) had the strongest correlation with the change in the CCS grade in the total study population (S2 Table). Changes in other dimensions were only weakly correlated with the CCS change (S2 Table). The correlations were similar across subgroups (Fig 2). The strongest individual correlation was observed in the PCI treatment group between the changes in the breathing dimension and the CCS grade

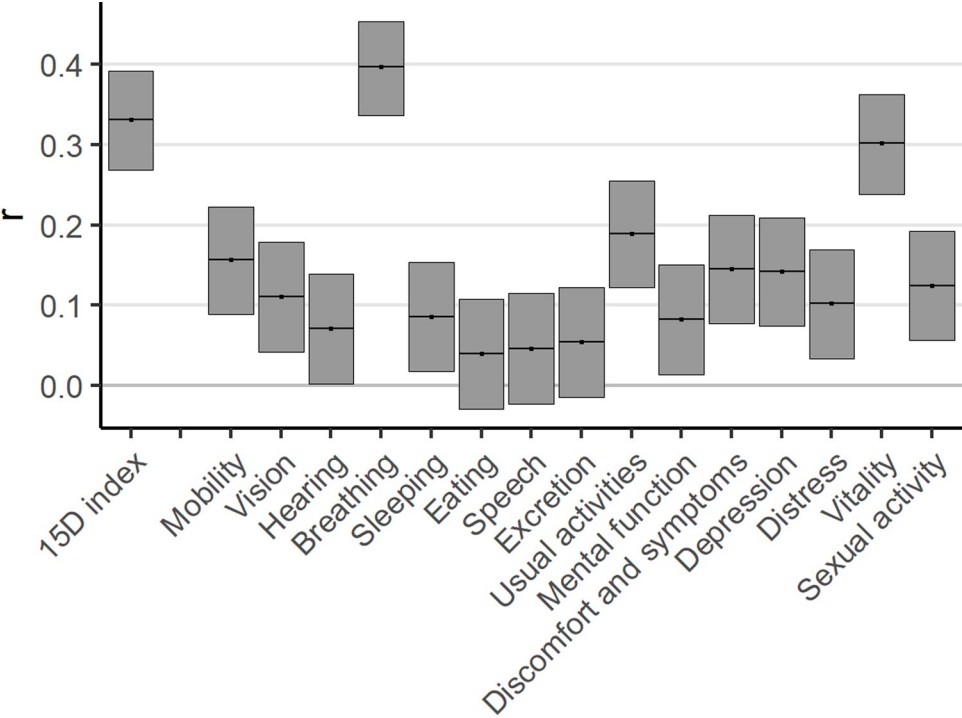

**Fig 1. Spearman correlation coefficients (95% confidence intervals) between one-year changes in the CCS and the 15D and its dimensions.**

(r = 0.46; 95% CI 0.36–0.55), but it did not differ statistically significantly from the other treatment groups (Fig 2A).

## Discussion

Our study shows that the patient-reported change in severity of CAD symptoms measured with CCS correlated moderately (r = 0.33) with the change in the generic HRQoL measured with the 15D. Strongest, although only moderate, correlations were observed between the change in the CCS grade and the changes in the 15D dimensions of breathing and vitality. The moderate correlation between the changes in the CCS grade and 15D index reflect that the CCS grade represents the severity of cardiac symptoms whereas the 15D is influenced, not only by cardiac health, but also by other dimensions of life, such as concomitant diseases [27]. These findings imply that symptom-based evaluation of the change in the CCS grade may not reflect the full benefit or harm of angina treatment, and thus, also other outcome measures are needed.

Although the correlation between CCS grade and 15D was only moderate, the proportion of patients with clinically meaningful improvement of symptoms was almost the same with both instruments; 47.2% of patients reported a clinically meaningful improvement in HRQoL measured with the 15D and 43.0% with the CCS. The correlation between CCS and 15D was weakened by the discrepancy in patients reporting worsening of symptoms or no benefit. Less than 10% of the patients reported increasing symptom severity (CCS grade), while every third patient reported worsening of generic HRQoL (15D). Correspondingly, using the CCS instrument, half of the patients did not experience any benefit, whereas this was the case on in only one fifth of patients when 15D was applied. This difference between CCS and 15D suggests that not only the severity of angina pectoris symptoms, but also other dimensions of health

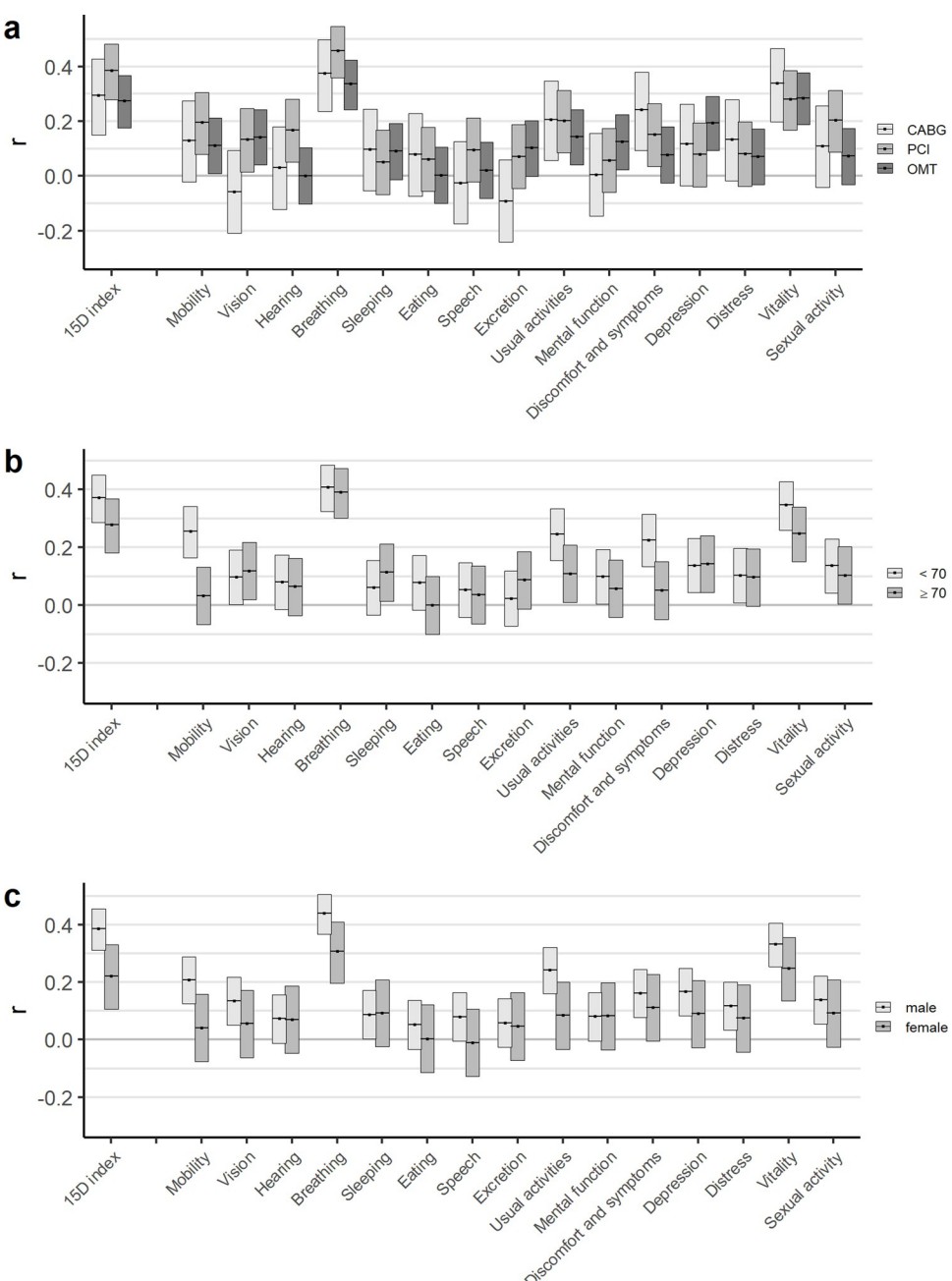

**Fig 2. Spearman correlation coefficients (95% confidence intervals) between one-year changes in the CCS and the 15D and its dimensions in subgroups.** (a) Correlations by treatment groups; (b) Correlations by age groups, <70 and ≥70; (c) Correlations by gender.

captured by the generic instrument are important aspects of quality of life. The proportion of CAD patients with deterioration of overall HRQoL was comparable with that reported in a previous study [28] demonstrating the external validity of our results.

In earlier studies, the association between CCS and NYHA class and generic instruments, such as EQ-5D and SF-6D have been somewhat controversial. In line with us, in the study by Goldsmith et al. one class increase of CCS grade, i.e. worsening of symptoms, was associated

with a decrease in HRQoL measured with the EQ-5D instrument among CAD and heart failure patients [17]. Similarly, in heart failure patients the EQ-5D and SF-6D instrument's utility scores decreased with increasing disease severity [16]. On the other hand, positive changes in EQ-5D and SF-6D instruments have been reported without co-occurring changes in the NYHA functional status [16]. This demonstrates that when assessing the overall therapeutic effect of a treatment, not only disease specific symptoms but also wider perspective of the patient's quality of life are needed.

With respect to the individual 15D dimensions, the CCS change correlated most strongly with changes in breathing (r = 0.40) and vitality (r = 0.30). The correlation between breathing and CCS was highest among the PCI patients (r = 0.46). This is expected based on the CCS grading definitions and is also in line with previous studies in CABG and PCI patients [28, 29]. Only weak correlations were observed between the change in the CCS grade and other dimensions of the 15D that would be expected to correlate with the CCS, including mobility and usual activities.

The prevalence of elderly patients with coronary artery disease is increasing. Elderly patients have greatest morbidity and mortality as well as high prevalence of comorbidities [30], which may influence the perceived effect of the treatment [31]. Indeed, in our study CCS class improved in 46% of patients in the younger age groups whereas only in 40% in the older age group. However, using the generic instrument (15D), the impact of age was less obvious; 15D improved in 48% and 46% in of the younger and the older population, respectively.

Routinely collected PRO data may not present the actual patient case-mix and may be biased towards younger and healthier patients [32]. A limitation of our study is that we did not have data on those patients who declined to answer the questionnaires at the baseline and, therefore, it is possible, that the studied patient population may not represent the actual case-mix of those treated. However, our study population is larger, and collected from a real-world setting which are strengths in comparison to previous smaller cross-sectional study [18] and randomized controlled trials [17].

## Conclusion

In conclusion, our cohort study provides real-life insight on the correlation between the changes in the CCS grade and the changes in the 15D in clinical practice. The observed moderate correlation between the changes in the CCS grade and the 15D indicates that the symptom-based evaluation of the change in the CCS grade may not catch the full benefit or harm of the treatment and vice versa, a generic instrument, such as 15D, may not capture the effect of treatment on symptoms. Although generic instruments, such as 15D, contain dimensions that to some extent capture changes in specific symptoms, disease-specific instruments are complementary and both are needed as they may be more sensitive for these changes. Therefore, both generic HRQoL and disease specific measurements are needed to capture the full effect of the treatment in future studies.

## Supporting information

**S1 Table. Spearman correlation coefficients (95% confidence limits) between change in the CCS and the 15D in different subgroups.**
(PDF)

**S2 Table. Spearman correlation coefficients (95% confidence limits) between change in the CCS and the 15D's dimensions.**
(PDF)

## Author Contributions

**Conceptualization:** Anna-Maija Tolppanen, Juha Hartikainen, Heikki Miettinen, Janne Martikainen, Risto P. Roine, Piia Lavikainen.

**Data curation:** Jarno Kotajärvi, Anna-Maija Tolppanen, Juha Hartikainen, Heikki Miettinen, Marketta Viljakainen, Risto P. Roine, Piia Lavikainen.

**Formal analysis:** Jarno Kotajärvi, Piia Lavikainen.

**Funding acquisition:** Anna-Maija Tolppanen, Juha Hartikainen, Heikki Miettinen, Janne Martikainen, Risto P. Roine.

**Investigation:** Marketta Viljakainen.

**Methodology:** Piia Lavikainen.

**Project administration:** Juha Hartikainen, Heikki Miettinen, Risto P. Roine.

**Resources:** Juha Hartikainen, Marketta Viljakainen.

**Software:** Piia Lavikainen.

**Supervision:** Anna-Maija Tolppanen, Juha Hartikainen, Heikki Miettinen, Janne Martikainen, Risto P. Roine, Piia Lavikainen.

**Visualization:** Jarno Kotajärvi.

**Writing – original draft:** Jarno Kotajärvi.

**Writing – review & editing:** Anna-Maija Tolppanen, Juha Hartikainen, Heikki Miettinen, Marketta Viljakainen, Janne Martikainen, Risto P. Roine, Piia Lavikainen.

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
