## [Decision Letter · Decision Letter 0]

2 Nov 2021

PONE-D-21-19920Correlation of the disease-specific Canadian Cardiovascular Society (CCS) classification and health-related quality of life (15D) in coronary artery disease patientsPLOS ONE

Dear Dr. Lavikainen,

Thank you for submitting your manuscript to PLOS ONE. After careful consideration, we feel that it has merit but does not fully meet PLOS ONE’s publication criteria as it currently stands. Therefore, we invite you to submit a revised version of the manuscript that addresses the points raised during the review process.

ACADEMIC EDITOR: Please address all comments from both referees.  Also, please rewrite lines 97-104 since those are the same sentences from the following article: Jari Heiskanen, Anna-Maija Tolppanen, Risto P Roine, Juha Hartikainen, Mikko Hippeläinen, Heikki Miettinen, Janne Martikainen, Comparison of EQ-5D and 15D instruments for assessing the health-related quality of life in cardiac surgery patients, European Heart Journal - Quality of Care and Clinical Outcomes, Volume 2, Issue 3, July 2016, Pages 193–200, https://doi.org/10.1093/ehjqcco/qcw002

We look forward to receiving your revised manuscript.

Kind regards,

Filipe Prazeres, MD, MSc, Ph.D.

Academic Editor

PLOS ONE

Journal Requirements:

2. Thank you for stating the following in the Competing Interests/Financial Disclosure* (delete as necessary) section:

“I have read the journal's policy and the authors of this manuscript have the following competing interests: Prof Martikainen is a founding partner of ESiOR Oy and a board member of Siltana Oy. These companies were not involved in carrying out this research. No other disclosures were reported”

We note that one or more of the authors are employed by a commercial company: name of commercial company.

Reviewers' comments:

Reviewer's Responses to Questions

**Comments to the Author**

1. Is the manuscript technically sound, and do the data support the conclusions?

Reviewer #1: Partly

Reviewer #2: Yes

2. Has the statistical analysis been performed appropriately and rigorously? 

Reviewer #1: No

Reviewer #2: No

3. Have the authors made all data underlying the findings in their manuscript fully available?

Reviewer #1: Yes

Reviewer #2: Yes

4. Is the manuscript presented in an intelligible fashion and written in standard English?

Reviewer #1: Yes

Reviewer #2: No

5. Review Comments to the Author

Reviewer #1: The manuscript aims to investigate the correlation between the change in patient-reported CCS and the generic 15D HRQoL score. The manuscript is structured but is regular written. The results present new and useful data for researchers and clinicians. The reviewer appreciates the effort of the authors. However, some parts need to be corrected. My specific comments are appended below.

Keywords:

-Rewrite “angina pectoris” as “Angina Pectoris” and “health-related quality of life” as “Health-Related Quality Of Life”. This is a MeSH term, and the first letter must be capitalized

-CCS and 15D are not a MeSH terms. I suggest you to change these terms as MeSH terms.

- I suggest you to keep keywords in alphabetical order

Introduction:

-In the first three paragraphs, the authors mention the importance of specific assessment of symptoms and HRQOL for economic analysis in health, but this is not the focus of the article, and there is no need to mention it repeatedly.

-I suggest the authors cite the study hypothesis.

Material and methods:

-I Suggest to separate ethical issues from the “Study design and population”. I suggest placing an item “Informed consent and ethical considerations” with the phrase of lines 89-91.

Results:

-It was not clear how the population was divided among those who completed each questionnaire. Since two instruments are being correlated, it is necessary for the studied population to complete the two questionnaires at both times.

-It would be important for the authors to explain the variation in participants, since 1,271 were included in the study, but different amounts filled each instrument (CCS and 15D).

-It would also be important to explain how many completed each instrument only at the first moment and why there were missing values and whether this would be a limitation for the study. Maybe na algorithm explain the included population would be helpfull.

-Suggestion line 148: put the percentage next to the number of patients

-S2 Table: Hearing dimension in the colun “95%” the autors wrote “0.00” I suggest change to “<0.01”

-The authors correlate HRQOL with procedures performed for angina (OMT, PCI, CABG), but sociodemographic characteristics (ie years of education, income) and clinical features (ie comorbidities, treatments for other diseases) affect HRQOL. It would be important to include this information in the results.

-Table 2: describe age ≥70 years, but they are a minority (45.5%) I suggest putting the majority

Discussion

-Line 195-196: The autors wrote 15D is influenced not only by heart health but also by other dimensions of life, such as concomitant illnesses. I suggest that in the results the authors describe whether the participants in this study had comorbidities i.e hypertension, diabetes.

-I suggest that the authors explain the criteria for selecting the age groups ≥70 / ≥ 70 years.

-I suggest discussing the results based on group age differences (≥70 / ≥ 70 years).

Reviewer #2: Methodology should be more clear.. and explained-well.

Under statistical analysis, only statistical methodology should be discussed/explained NOT the overall/general scoring etc etc.

Discussion should be re-written/revised appropriately matching with the objectives and the findings obtained, not just like expanding the intro part.

all the obtained results must be discussed in detail and by referring to/with earlier studies using same/similar research tools.

6. PLOS authors have the option to publish the peer review history of their article (what does this mean?). If published, this will include your full peer review and any attached files.

Reviewer #1: **Yes: **Priscila Moreno Sperling Cannavan

Reviewer #2: No

---

## [Author Response · Author response to Decision Letter 0]

17 Dec 2021

Response to Reviewers

Comments to the Author

1. Is the manuscript technically sound, and do the data support the conclusions?

Reviewer #1: Partly

Reviewer #2: Yes

2. Has the statistical analysis been performed appropriately and rigorously?

Reviewer #1: No

Reviewer #2: No

3. Have the authors made all data underlying the findings in their manuscript fully available?

Reviewer #1: Yes

Reviewer #2: Yes

4. Is the manuscript presented in an intelligible fashion and written in standard English?

Reviewer #1: Yes

Reviewer #2: No

5. Review Comments to the Author

Reviewer #1: The manuscript aims to investigate the correlation between the change in patient-reported CCS and the generic 15D HRQoL score. The manuscript is structured but is regular written. The results present new and useful data for researchers and clinicians. The reviewer appreciates the effort of the authors. However, some parts need to be corrected. My specific comments are appended below.

Keywords:

-Rewrite “angina pectoris” as “Angina Pectoris” and “health-related quality of life” as “Health-Related Quality Of Life”. This is a MeSH term, and the first letter must be capitalized

-CCS and 15D are not a MeSH terms. I suggest you to change these terms as MeSH terms.

- I suggest you to keep keywords in alphabetical order

Our response: We thank the Reviewer for commenting on this. We have rewritten the terms according to MeSH terms. We would prefer to include also 15D and CCS as key words although they are not MeSH terms but leave the final decision on this to the Editor. We have also sorted the keywords to alphabetical order.

Introduction:

-In the first three paragraphs, the authors mention the importance of specific assessment of symptoms and HRQOL for economic analysis in health, but this is not the focus of the article, and there is no need to mention it repeatedly.

Our response: We agree with the Reviewer and have modified the introduction so that health economics analyses are emphasized less.

-I suggest the authors cite the study hypothesis.

Our response: We would rather refrain from stating an exact hypothesis, because we did not have an a priori hypothesis for what we expected to find. Our aim was to assess whether the change measured by these two patient-reported instruments is correlated. However, we have revised the last paragraph of the introduction so that the rationale for our study is stated more clearly. The revised paragraph reads as follows: “Previous studies have assessed the correlation between clinician measured CCS grading or New York Heart Association (NYHA) functional classification and patient-reported generic HRQoL, mainly in cross-sectional designs [16–18]. Therefore, it is unknown whether the change in symptom severity evaluated by the CCS is associated with the change in generic HRQoL. Further, the earlier studies have usually compared physician-evaluated CCS to a patient-reported generic HRQoL instrument. Therefore, it is unknow whether the change in self-reported CCS correlates with the change in self-reported generic HRQoL. To evaluate this, we investigated the correlation between the change in patient-reported CCS and the generic 15D HRQoL score among patients with coronary artery disease.”

Material and methods:

-I Suggest to separate ethical issues from the “Study design and population”. I suggest placing an item “Informed consent and ethical considerations” with the phrase of lines 89-91.

Our response: In accordance with the suggestion of the Reviewer, we have separated ethical issues to a new paragraph with a heading “Informed consent and ethical considerations”. 

Results:

-It was not clear how the population was divided among those who completed each questionnaire. Since two instruments are being correlated, it is necessary for the studied population to complete the two questionnaires at both times.

Our response: We agree with the Reviewer and have clarified the number of people in the correlation analysis in the revised manuscript (lines183-184) as follows: “Altogether 805 (63.3%) patients had both CCS and 15D index scores reported at both timepoints and were included in these analyses.” We have also specified the number of patients included in the analyses assessing the changes in individual instruments on lines 167-168 and 174-175.

-It would be important for the authors to explain the variation in participants, since 1,271 were included in the study, but different amounts filled each instrument (CCS and 15D).

Our response: Please, see our answer to the previous comment. 

-It would also be important to explain how many completed each instrument only at the first moment and why there were missing values and whether this would be a limitation for the study. Maybe na algorithm explain the included population would be helpfull.

Our response: We have reported the number (proportion) of missing data at baseline and after 12 months for both instruments in table 2. In addition, the number of patients with data available at both timepoints is reported in the table and Results section of the manuscript (lines 167-168, 174-175 and 183-184). Unfortunately, we have no data on the reasons for missing data. 

-Suggestion line 148: put the percentage next to the number of patients

Our response: In line with the recommendation of the Reviewer, we have added the proportions next to the number of patients on lines 167, 174 and 183.

-S2 Table: Hearing dimension in the colun “95%” the autors wrote “0.00” I suggest change to “<0.01”

Our response: In accordance with the suggestion of the Reviewer, we have corrected the correlation coefficient as “<0.01”.

-The authors correlate HRQOL with procedures performed for angina (OMT, PCI, CABG), but sociodemographic characteristics (ie years of education, income) and clinical features (ie comorbidities, treatments for other diseases) affect HRQOL. It would be important to include this information in the results.

Our response: We thank the Reviewer for the comment. Unfortunately, we have no data on sociodemographic characteristics or clinical features.

-Table 2: describe age ≥70 years, but they are a minority (45.5%) I suggest putting the majority

Our response: In line with the recommendation of the Reviewer, we have changed the age to describe the majority, i.e., patients <70 years.

Discussion

-Line 195-196: The autors wrote 15D is influenced not only by heart health but also by other dimensions of life, such as concomitant illnesses. I suggest that in the results the authors describe whether the participants in this study had comorbidities i.e hypertension, diabetes.

Our response: We agree that describing the comorbidities would inform on the comorbidities of the study population, but we would prefer to keep the tables and manuscript in the current form. The main aim of our manuscript was to illustrate whether the changes in symptom-specific CCS and generic 15D instrument are consistently correlated (i.e., how much they measure the same phenomenon, can 15D capture the patient-reported change in CCS). As all participants had coronary artery disease, most of them had hypertension and/or diabetes and thus reporting the prevalence of these comorbidities would not result to a meaningful improvement in the manuscript. 

-I suggest that the authors explain the criteria for selecting the age groups ≥70 / ≥ 70 years.

Our response: The aim was to study the treatment effect in two age groups (young and old) with approximately equal size (quantiles). As the median age in the study population (n=1271) was 69 years, rounding it to the nearest tenth resulted in two age groups (<70 and ≥70 years) with approximately same size (54.5 % and 45.5% of the study population).

-I suggest discussing the results based on group age differences (≥70 / ≥ 70 years).

Our response: In accordance with the suggestion of the Reviewer, we have added to the discussion section as follows (lines 267-272): ”The prevalence of elderly patients with coronary artery disease is increasing. Elderly patients have greatest morbidity and mortality as well as high prevalence of comorbidities [30], which may influence the perceived effect of treatment [31]. Indeed, in our study CCS class improved in 46% of patients in the younger age groups whereas only in 40% in the older age group. However, using the generic instrument (15D), the impact of age was less obvious; 15D improved in 48% and 46% in of the younger and the older population, respectively.“

Reviewer #2: Methodology should be more clear.. and explained-well.

Under statistical analysis, only statistical methodology should be discussed/explained NOT the overall/general scoring etc etc.

Our response: We thank the Reviewer for the comment. We have restructured the section of statistical analyses according to the suggestion of the Reviewer.

Discussion should be re-written/revised appropriately matching with the objectives and the findings obtained, not just like expanding the intro part. all the obtained results must be discussed in detail and by referring to/with earlier studies using same/similar research tools.

Our response: 

We thank the Reviewer for the suggestion and hope that the modification of the Introduction section as suggested by the Reviewer 1 above has removed some of the overlap. 

We also agree that the Discussion to some extent “expands” the Introduction as the aim of the Introduction is to provide the context for performing the study, including a brief introduction to earlier studies related to the topic. We have rewritten and reorganized the Discussion in the revised manuscript. We hope that in the revised form the discussion of our results in the context of earlier studies on this topic is more evident. The following text has been added to the Discussion:

“Although the correlation between CCS grade and 15D was only moderate, the proportion of patients with clinically meaningful improvement of symptoms was almost the same with both instruments; 47.2% of patients reported a clinically meaningful improvement in HRQoL measured with the 15D and 43.0% with the CCS. The correlation between CCS and 15D was weakened by the discrepancy in patients reporting worsening of symptoms or no benefit. Less than 10% of the patients reported increasing symptom severity (CCS grade), while every third patient reported worsening of generic HRQoL (15D). Correspondingly, using the CCS instrument, half of the patients did not experience any benefit, whereas this was the case on in only one fifth of patients when 15D was applied. This difference between CCS and 15D suggests that not only the severity of angina pectoris symptoms, but also other dimensions of health captured by the generic instrument are important aspects of quality of life. The proportion of CAD patients with deterioration of overall HRQoL was comparable with that reported in a previous study [28] demonstrating the external validity of our results.

In earlier studies, the association between CCS and NYHA class and generic instruments, such as EQ-5D and SF-6D have been somewhat controversial. In line with us, in the study by Goldsmith et al. one class increase of CCS grade, i.e. worsening of symptoms, was associated with a decrease in HRQoL measured with the EQ-5D instrument among CAD and heart failure patients [17]. Similarly, in heart failure patients the EQ-5D and SF-6D instrument’s utility scores decreased with increasing disease severity [16]. On the other hand, positive changes in EQ-5D and SF-6D instruments have been reported without co-occurring changes in the NYHA functional status [16]. This demonstrates that when assessing the overall therapeutic effect of a treatment, not only disease specific symptoms but also wider perspective of the patient’s quality of life are needed.”

6. PLOS authors have the option to publish the peer review history of their article (what does this mean?). If published, this will include your full peer review and any attached files.

Do you want your identity to be public for this peer review? For information about this choice, including consent withdrawal, please see our Privacy Policy.

Reviewer #1: Yes: Priscila Moreno Sperling Cannavan

Reviewer #2: No

---

## [Decision Letter · Decision Letter 1]

9 Feb 2022

PONE-D-21-19920R1Correlation of the disease-specific Canadian Cardiovascular Society (CCS) classification and health-related quality of life (15D) in coronary artery disease patientsPLOS ONE

Dear Dr. Lavikainen,

Thank you for submitting your manuscript to PLOS ONE. After careful consideration, we feel that it has merit but does not fully meet PLOS ONE’s publication criteria as it currently stands. Therefore, we invite you to submit a revised version of the manuscript that addresses the points raised during the review process.

I also suggest that the authors review the order of the references throughout the manuscript. Some paragraphs were redacted, but the reference number did not change. 

We look forward to receiving your revised manuscript.

Kind regards,

Filipe Prazeres, MD, MSc, Ph.D.

Academic Editor

PLOS ONE

Journal Requirements:

Reviewers' comments:

Reviewer's Responses to Questions

**Comments to the Author**

1. If the authors have adequately addressed your comments raised in a previous round of review and you feel that this manuscript is now acceptable for publication, you may indicate that here to bypass the “Comments to the Author” section, enter your conflict of interest statement in the “Confidential to Editor” section, and submit your "Accept" recommendation.

Reviewer #1: (No Response)

Reviewer #2: All comments have been addressed

2. Is the manuscript technically sound, and do the data support the conclusions?

Reviewer #1: (No Response)

Reviewer #2: Partly

3. Has the statistical analysis been performed appropriately and rigorously? 

Reviewer #1: (No Response)

Reviewer #2: Yes

4. Have the authors made all data underlying the findings in their manuscript fully available?

Reviewer #1: (No Response)

Reviewer #2: Yes

5. Is the manuscript presented in an intelligible fashion and written in standard English?

Reviewer #1: (No Response)

Reviewer #2: Yes

6. Review Comments to the Author

Reviewer #1: (No Response)

Reviewer #2: They have revised it, and in this form it could be accepted. Appreciated the revised version. May proceed

7. PLOS authors have the option to publish the peer review history of their article (what does this mean?). If published, this will include your full peer review and any attached files.

Reviewer #1: No

Reviewer #2: No

---

## [Author Response · Author response to Decision Letter 1]

14 Feb 2022

Response to Reviewers:

1- The authors made several corrections and as the manuscript was sent for a new evaluation, there is no way to evaluate it, because the corrections and the first version of the manuscript are together in the same version, the correction with marking (ie: manuscript), leaving it confusing for the correction.

I suggest a new submission, only with the second version of the manuscript and that the authors highlight what was changed (i.e: manuscript).

Our response: We apologize for the inconvenience. However, the revised manuscript is submitted with tracked changes (utilizing “Track changes” tool in Microsoft Word) as suggested by the journal. In addition, the submission includes a clean version without tracked changes.

2- I also suggest that authors review the order of references in the introduction. The paragraphs were redacted, but the reference number did not change.

Our response: We are grateful for the Reviewer for noticing this. We have reviewed and corrected the order of the references.

---

## [Decision Letter · Decision Letter 2]

15 Mar 2022

Correlation of the disease-specific Canadian Cardiovascular Society (CCS) classification and health-related quality of life (15D) in coronary artery disease patients

PONE-D-21-19920R2

Dear Dr. Lavikainen,

We’re pleased to inform you that your manuscript has been judged scientifically suitable for publication and will be formally accepted for publication once it meets all outstanding technical requirements.

Kind regards,

Filipe Prazeres, MD, MSc, Ph.D.

Academic Editor

PLOS ONE

Additional Editor Comments (optional):

Reviewers' comments:

Reviewer's Responses to Questions

**Comments to the Author**

1. If the authors have adequately addressed your comments raised in a previous round of review and you feel that this manuscript is now acceptable for publication, you may indicate that here to bypass the “Comments to the Author” section, enter your conflict of interest statement in the “Confidential to Editor” section, and submit your "Accept" recommendation.

Reviewer #1: All comments have been addressed

2. Is the manuscript technically sound, and do the data support the conclusions?

Reviewer #1: Yes

3. Has the statistical analysis been performed appropriately and rigorously? 

Reviewer #1: Yes

4. Have the authors made all data underlying the findings in their manuscript fully available?

Reviewer #1: Yes

5. Is the manuscript presented in an intelligible fashion and written in standard English?

Reviewer #1: Yes

6. Review Comments to the Author

Reviewer #1: The authors have reviewed the manuscript and so it can be accepted and published.

Appreciated the revised version.

7. PLOS authors have the option to publish the peer review history of their article (what does this mean?). If published, this will include your full peer review and any attached files.

Reviewer #1: **Yes: **Priscila Moreno Sperling Cannavan

---

## [Editor Report · Acceptance letter]

25 Mar 2022

PONE-D-21-19920R2 

Correlation of the disease-specific Canadian Cardiovascular Society (CCS) classification and health-related quality of life (15D) in coronary artery disease patients 

Dear Dr. Lavikainen:

I'm pleased to inform you that your manuscript has been deemed suitable for publication in PLOS ONE. Congratulations! Your manuscript is now with our production department. 

Kind regards, 

on behalf of

Prof. Filipe Prazeres 

Academic Editor

PLOS ONE